# Recurrent Neural Networks for Multivariate Time Series with Missing Values

**Zhengping Che, Sanjay Purushotham**
Department of Computer Science
University of Southern California
Los Angeles, CA 90089, USA
`{zche,spurusho}@usc.edu`

**Kyunghyun Cho, David Sontag**
Department of Computer Science
New York University
New York, NY 10012, USA
`kyunghyun.cho@nyu.edu,dsontag@cs.nyu.edu`

**Yan Liu**
Department of Computer Science
University of Southern California
Los Angeles, CA 90089, USA
`yanliu.cs@usc.edu`

## Abstract

Multivariate time series data in practical applications, such as health care, geoscience, and biology, are characterized by a variety of missing values. In time series prediction and other related tasks, it has been noted that missing values and their missing patterns are often correlated with the target labels, a.k.a., *informative* missingness. There is very limited work on exploiting the missing patterns for effective imputation and improving prediction performance. In this paper, we develop novel deep learning models, namely GRU-D, as one of the early attempts. GRU-D is based on Gated Recurrent Unit (GRU), a state-of-the-art recurrent neural network. It takes two representations of missing patterns, i.e., *masking* and *time interval*, and effectively incorporates them into a deep model architecture so that it not only captures the long-term temporal dependencies in time series, but also utilizes the missing patterns to achieve better prediction results. Experiments of time series classification tasks on real-world clinical datasets (MIMIC-III, PhysioNet) and synthetic datasets demonstrate that our models achieve state-of-the-art performance and provides useful insights for better understanding and utilization of missing values in time series analysis.

## 1    Introduction

Multivariate time series data are ubiquitous in many practical applications ranging from health care, geoscience, astronomy, to biology and others. They often inevitably carry missing observations due to various reasons, such as medical events, saving costs, anomalies, inconvenience and so on. It has been noted that these missing values are usually *informative missingness* (Rubin, 1976), i.e., the missing values and patterns provide rich information about target labels in supervised learning tasks (e.g, time series classification). To illustrate this idea, we show some examples from MIMIC-III, a real world health care dataset in Figure 1. We plot the Pearson correlation coefficient between variable missing rates, which indicates how often the variable is missing in the time series, and the labels of our interests such as mortality and ICD-9 diagnoses. We observe that the missing rate is correlated with the labels, and the missing rates with low rate values are usually highly (either positive or negative) correlated with the labels. These findings demonstrate the usefulness of missingness patterns in solving a prediction task.

In the past decades, various approaches have been developed to address missing values in time series (Schafer & Graham, 2002). A simple solution is to omit the missing data and to perform analysis only on the observed data. A variety of methods have been developed to fill in the missing values, such as smoothing or interpolation (Kreindler & Lumsden, 2012), spectral analysis (Mondal & Percival, 2010), kernel methods (Rehfeld et al., 2011), multiple imputation (White et al., 2011),

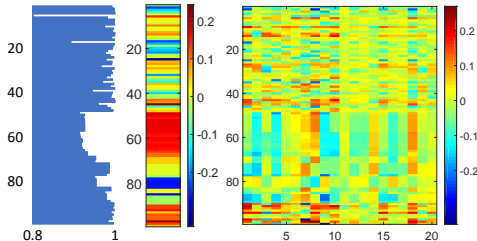

Figure 1: Demonstrations of informative missingness on MIMIC-III dataset. Left figure shows variable missing rate (x-axis, missing rate; y-axis, input variable). Middle/right figures respectively shows the correlations between missing rate and mortality/ICD-9 diagnosis categories (x-axis, target label; y-axis, input variable; color, correlation value). Please refer to Appendix A.1 for more details.

and EM algorithm (García-Laencina et al., 2010). Schafer & Graham (2002) and references therein provide excellent reviews on related solutions. However, these solutions often result in a two-step process where imputations are disparate from prediction models and missing patterns are not effectively explored, thus leading to suboptimal analyses and predictions (Wells et al., 2013).

In the meantime, Recurrent Neural Networks (RNNs), such as Long Short-Term Memory (LSTM) (Hochreiter & Schmidhuber, 1997) and Gated Recurrent Unit (GRU) (Cho et al., 2014), have shown to achieve the state-of-the-art results in many applications with time series or sequential data, including machine translation (Bahdanau et al., 2014; Sutskever et al., 2014) and speech recognition (Hinton et al., 2012). RNNs enjoy several nice properties such as strong prediction performance as well as the ability to capture long-term temporal dependencies and variable-length observations. RNNs for missing data has been studied in earlier works (Bengio & Gingras, 1996; Tresp & Briegel, 1998; Parveen & Green, 2001) and applied for speech recognition and blood-glucose prediction. Recent works (Lipton et al., 2016; Choi et al., 2015) tried to handle missingness in RNNs by concatenating missing entries or timestamps with the input or performing simple imputations. However, there have not been works which model missing patterns into a systematically modified RNN structure for time series classification problems. Exploiting the power of customized RNN models along with the *informativeness* of missing patterns is a new promising venue to effectively model multivariate time series and is the main motivation behind our work.

In this paper, we develop a novel deep learning model based on GRU, namely GRU-D, to effectively exploit two representations of informative missingness patterns, i.e., *masking* and *time interval*. Masking informs the model which inputs are observed (or missing), while time interval encapsulates the input observation patterns. Our model captures the observations and their dependencies by applying masking and time interval (using a decay term) to the inputs and network states of GRU, and jointly train all model components using back-propagation. Thus, our model not only captures the long-term temporal dependencies of time series observations but also utilizes the missing patterns to improve the prediction results. Empirical experiments on real-world clinical datasets as well as synthetic datasets demonstrate that our proposed model outperforms strong deep learning models built on GRU with imputation as well as other strong baselines. These experiments show that our proposed method is suitable for many time series classification problems with missing data, and in particular is readily applicable to the predictive tasks in emerging health care applications. Moreover, our method provides useful insights into more general research challenges of time series analysis with missing data beyond classification tasks, including 1) a general deep learning framework to handle time series with missing data, 2) effective solutions to characterize the missing patterns of not missing-completely-at-random time series data such as modeling masking and time interval, and 3) an insightful approach to study the impact of variable missingness on the prediction labels by decay analysis.

## 2 RNN MODELS FOR TIME SERIES WITH MISSING VARIABLES

We denote a multivariate time series with $D$ variables of length $T$ as $\boldsymbol{X} = (\boldsymbol{x}_1, \boldsymbol{x}_2, \ldots, \boldsymbol{x}_T)^{\mathrm{T}} \in \mathbb{R}^{T \times D}$, where for each $t \in \{1, 2, \ldots, T\}$, $\boldsymbol{x}_t \in \mathbb{R}^D$ represents the $t$th observations (a.k.a., measurements) of all variables and $x_t^d$ denotes the measurement of $d$th variable of $\boldsymbol{x}_t$. Let $s_t \in \mathbb{R}$ denote the time-stamp when the $t$th observation is obtained and we assume that the first observation is made at

$\boldsymbol{X}$: Input time series (2 variables); $\boldsymbol{M}$: Masking for $\boldsymbol{X}$;

$\boldsymbol{s}$: Timestamps for $\boldsymbol{X}$; $\boldsymbol{\Delta}$: Time interval for $\boldsymbol{X}$.

$$\boldsymbol{X} = \begin{bmatrix} 47 & 49 & NA & 40 & NA & 43 & 55 \\ NA & 15 & 14 & NA & NA & NA & 15 \end{bmatrix} \quad \boldsymbol{M} = \begin{bmatrix} 1 & 1 & 0 & 1 & 0 & 1 & 1 \\ 0 & 1 & 1 & 0 & 0 & 0 & 1 \end{bmatrix}$$

$$\boldsymbol{s} = \begin{bmatrix} 0 & 0.1 & 0.6 & 1.6 & 2.2 & 2.5 & 3.1 \end{bmatrix} \quad \boldsymbol{\Delta} = \begin{bmatrix} 0.0 & 0.1 & 0.5 & 1.5 & 0.6 & 0.9 & 0.6 \\ 0.0 & 0.1 & 0.5 & 1.0 & 1.6 & 1.9 & 2.5 \end{bmatrix}$$

Figure 2: An example of measurement vectors $\boldsymbol{x}_t$, time stamps $s_t$, masking $\boldsymbol{m}_t$, and time interval $\boldsymbol{\delta}_t$.

time-stamp 0 (i.e., $s_1 = 0$). A time series $\boldsymbol{X}$ could have missing values. We introduce a *masking vector* $\boldsymbol{m}_t \in \{0, 1\}^D$ to denote which variables are missing at time step $t$. The masking vector for $\boldsymbol{x}_t$ is given by

$$m_t^d = \begin{cases} 1, & \text{if } x_t^d \text{ is observed} \\ 0, & \text{otherwise} \end{cases}$$

For each variable $d$, we also maintain the *time interval* $\delta_t^d \in \mathbb{R}$ since its last observation as

$$\delta_t^d = \begin{cases} s_t - s_{t-1} + \delta_{t-1}^d, & t > 1, m_{t-1}^d = 0 \\ s_t - s_{t-1}, & t > 1, m_{t-1}^d = 1 \\ 0, & t = 1 \end{cases}$$

An example of these notations is illustrated in Figure 2. In this paper, we are interested in the time series classification problem, where we predict the labels $l_n$ given the time series data $\mathcal{D}$, where $\mathcal{D} = \{(\boldsymbol{X}_n, \boldsymbol{s}_n, \boldsymbol{M}_n, \Delta_n)\}_{n=1}^{N}$, and $\boldsymbol{X}_n = \left[ \boldsymbol{x}_1^{(n)}, \ldots, \boldsymbol{x}_{T_n}^{(n)} \right]$, $\boldsymbol{s}_n = \left[ s_1^{(n)}, \ldots, s_{T_n}^{(n)} \right]$, $\boldsymbol{M}_n = \left[ \boldsymbol{m}_1^{(n)}, \ldots, \boldsymbol{m}_{T_n}^{(n)} \right]$, $\Delta_n = \left[ \boldsymbol{\delta}_1^{(n)}, \ldots, \boldsymbol{\delta}_{T_n}^{(n)} \right]$, and $l_n \in \{1, \ldots, L\}$.

## 2.1 GRU-RNN FOR TIME SERIES CLASSIFICATION

We investigate the use of recurrent neural networks (RNN) for time-series classification, as their recursive formulation allow them to handle variable-length sequences naturally. Moreover, RNN shares the same parameters across all time steps which greatly reduces the total number of parameters we need to learn. Among different variants of the RNN, we specifically consider an RNN with gated recurrent units (Cho et al., 2014; Chung et al., 2014), but similar discussion and convolutions are also valid for other RNN models such as LSTM (Hochreiter & Schmidhuber, 1997).

The structure of GRU is shown in Figure 3(a). For each $j$th hidden unit, GRU has a reset gate $r_t^j$ and an update gate $z_t^j$ to control the hidden state $h_t^j$ at each time $t$. The update functions are shown as follows:

$$\begin{aligned} \boldsymbol{r}_t &= \sigma \left( \boldsymbol{W}_r \boldsymbol{x}_t + \boldsymbol{U}_r \boldsymbol{h}_{t-1} + \boldsymbol{b}_r \right) & \boldsymbol{z}_t &= \sigma \left( \boldsymbol{W}_z \boldsymbol{x}_t + \boldsymbol{U}_z \boldsymbol{h}_{t-1} + \boldsymbol{b}_z \right) \\ \tilde{\boldsymbol{h}}_t &= \tanh \left( \boldsymbol{W} \boldsymbol{x}_t + \boldsymbol{U}(\boldsymbol{r}_t \odot \boldsymbol{h}_{t-1}) + \boldsymbol{b} \right) & \boldsymbol{h}_t &= (\boldsymbol{1} - \boldsymbol{z}_t) \odot \boldsymbol{h}_{t-1} + \boldsymbol{z}_t \odot \tilde{\boldsymbol{h}}_t \end{aligned}$$

where matrices $\boldsymbol{W}_z, \boldsymbol{W}_r, \boldsymbol{W}, \boldsymbol{U}_z, \boldsymbol{U}_r, \boldsymbol{U}$ and vectors $\boldsymbol{b}_z, \boldsymbol{b}_r, \boldsymbol{b}$ are model parameters. We use $\sigma$ for element-wise sigmoid function, and $\odot$ for element-wise multiplication. This formulation assumes that all the variables are observed. A sigmoid or soft-max layer is then applied on the output of the GRU layer at the last time step for classification task.

Existing work on handling missing values lead to three possible solutions with no modification on GRU network structure. One straightforward approach is simply replacing each missing observation

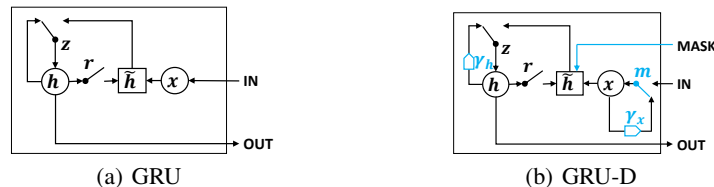

(a) GRU (b) GRU-D

Figure 3: Graphical illustrations of the original GRU (left) and the proposed GRU-D (right) models.

with the mean of the variable across the training examples. In the context of GRU, we have

$$x_t^d \leftarrow m_t^d x_t^d + (1 - m_t^d) \tilde{x}^d \tag{1}$$

where $\tilde{x}^d = \sum_{n=1}^{N} \sum_{t=1}^{T_n} m_{t,n}^d x_{t,n}^d \Big/ \sum_{n=1}^{N} \sum_{t=1}^{T_n} m_{t,n}^d$. We refer to this approach as **GRU-mean**.

A second approach is exploiting the temporal structure in time series. For example, we may assume any missing value is same as its last measurement and use forward imputation (**GRU-forward**), i.e.,

$$x_t^d \leftarrow m_t^d x_t^d + (1 - m_t^d) x_{t'}^d \tag{2}$$

where $t' < t$ is the last time the $d$-th variable was observed.

Instead of explicitly imputing missing values, the third approach simply indicates which variables are missing and how long they have been missing as a part of input, by concatenating the measurement, masking and time interval vectors as

$$\boldsymbol{x}_t^{(n)} \leftarrow \left[ \boldsymbol{x}_t^{(n)}; \boldsymbol{m}_t^{(n)}; \boldsymbol{\delta}_t^{(n)} \right] \tag{3}$$

where $\boldsymbol{x}_t^{(n)}$ can be either from Equation (1) or (2). We later refer to this approach as **GRU-simple**.

Several recent works (Lipton et al., 2016; Choi et al., 2015; Pham et al., 2016) use RNNs on EHR data to model diseases and to predict patient diagnosis from health care time series data with irregular time stamps or missing values, but none of them have explicitly attempted to capture and utilize the missing patterns into their RNNs via systematically modified network architectures. Choi et al. (2015) feeds medical codes along with its time stamps into GRU model to predict the next medical event. This feeding time stamps idea is equivalent to the baseline GRU-simple without feeding the masking, which we denote as **GRU-simple (interval only)**. Pham et al. (2016) takes time stamps into LSTM model, and modify its forgetting gate by either time decay and parametric time both from time stamps. However, their non-trainable decay is not that flexible, and the parametric time also does not change RNN model structure and is similar to GRU-simple (interval only). In addition, neither of them consider missing values in time series medical records, and the time stamp input used in these two models is vector for one patient, but not matrix for each input variable of one patient as ours. Lipton et al. (2016) achieves their best performance on diagnosis prediction by feeding masking with zero-filled missing values. Their model is equivalent to GRU-simple without feeding the time interval, and no model structure modification is made for further capturing and utilizing missingness. We denote their best model as **GRU-simple (masking only)**. Conclusively, our GRU-simple baseline can be considered as a generalization from all related RNN models mentioned above and as shown in the experiments these GRU-simple variations have quite close performance.

These approaches solve the missing value issue to a certain extent, However, it is known that imputing the missing value with mean or forward imputation cannot distinguish whether missing values are imputed or truly observed. Simply concatenating masking and time interval vectors fails to exploit the temporal structure of missing values. Thus none of them fully utilize missingness in data to achieve desirable performance.

## 2.2 GRU-D: MODEL WITH TRAINABLE DECAYS

To fundamentally address the issue of missing values in time series, we notice two important properties of the missing values in time series, especially in health care domains: First, the value of the missing variable tend to be close to some default value if its last observation happens a long time ago. This property usually exists in health care data for human body as homeostasis mechanisms and is considered to be critical for disease diagnosis and treatment (Vodovotz et al., 2013). Second, the influence of the input variables will fade away over time if the variable has been missing for a while. For example, one medical feature in electronic health records (EHRs) is only significant in a certain temporal context (Zhou & Hripcsak, 2007). Therefore we propose a GRU-based model called **GRU-D**, in which a *decay* mechanism is designed for the input variables and the hidden states to capture the aforementioned properties. We introduce *decay rates* in the model to control the decay mechanism by considering the following important factors. First, each input variable in health care time series has its own medical meaning and importance. The decay rates should be flexible to differ from variable to variable based on the underlying properties associated with the variables.

Second, as we see lots of missing patterns are informative in prediction tasks, the decay rate should be indicative of such patterns and benefits the prediction tasks. Furthermore, since the missing patterns are unknown and possibly complex, we aim at learning decay rates from the training data rather than being fixed a priori. That is, we model a vector of decay rates $\boldsymbol{\gamma}$ as

$$\boldsymbol{\gamma}_t = \exp\left\{-\max\left(\mathbf{0}, \boldsymbol{W}_\gamma \boldsymbol{\delta_t} + \boldsymbol{b}_\gamma\right)\right\} \tag{4}$$

where $\boldsymbol{W}_\gamma$ and $\boldsymbol{b}_\gamma$ are model parameters that we train jointly with all the other parameters of the GRU. We chose the exponentiated negative rectifier in order to keep each decay rate monotonically decreasing in a reasonable range between $0$ and $1$. Note that other formulations such as a sigmoid function can be used instead, as long as the resulting decay is monotonic and is in the same range.

Our proposed **GRU-D** model incorporates two different trainable decays to utilize the missingness directly with the input feature values and implicitly in the RNN states. First, for a missing variable, we use an *input decay* $\boldsymbol{\gamma}_{\boldsymbol{x}}$ to decay it over time toward the empirical mean (which we take as a *default* configuration), instead of using the last observation as it is. Under this assumption, the trainable decay scheme can be readily applied to the measurement vector by

$$x_t^d \leftarrow m_t^d x_t^d + (1 - m_t^d)\gamma_{\boldsymbol{x}_t}^d x_{t'}^d + (1 - m_t^d)(1 - \gamma_{\boldsymbol{x}_t}^d)\tilde{x}^d \tag{5}$$

where $x_{t'}^d$ is the last observation of the $d$-th variable ($t' < t$) and $\tilde{x}^d$ is the empirical mean of the $d$th variable. When decaying the input variable directly, we constrain $\boldsymbol{W}_{\gamma_{\boldsymbol{x}}}$ to be diagonal, which effectively makes the decay rate of each variable independent from the others. Sometimes the input decay may not fully capture the missing patterns since not all missingness information can be represented in decayed input values. In order to capture richer knowledge from missingness, we also have a *hidden state decay* $\boldsymbol{\gamma_h}$ in GRU-D. Intuitively, this has an effect of decaying the extracted features (GRU hidden states) rather than raw input variables directly. This is implemented by decaying the previous hidden state $\boldsymbol{h}_{t-1}$ before computing the new hidden state $\boldsymbol{h}_t$ as

$$\boldsymbol{h}_{t-1} \leftarrow \boldsymbol{\gamma_{ht}} \odot \boldsymbol{h}_{t-1}, \tag{6}$$

in which case we do not constrain $\boldsymbol{W}_{\gamma_h}$ to be diagonal. In addition, we feed the masking vectors ($\boldsymbol{m}_t$) directly into the model. The update functions of GRU-D are

$$\boldsymbol{z}_t = \sigma\left(\boldsymbol{W}_z\boldsymbol{x}_t + \boldsymbol{U}_z\boldsymbol{h}_{t-1} + \boldsymbol{V}_z\boldsymbol{m}_t + \boldsymbol{b}_z\right) \qquad \boldsymbol{r}_t = \sigma\left(\boldsymbol{W}_r\boldsymbol{x}_t + \boldsymbol{U}_r\boldsymbol{h}_{t-1} + \boldsymbol{V}_r\boldsymbol{m}_t + \boldsymbol{b}_r\right)$$
$$\tilde{\boldsymbol{h}}_t = \tanh\left(\boldsymbol{W}\boldsymbol{x}_t + \boldsymbol{U}(\boldsymbol{r}_t \odot \boldsymbol{h}_{t-1}) + \boldsymbol{V}\boldsymbol{m}_t + \boldsymbol{b}\right) \quad \boldsymbol{h}_t = (\mathbf{1} - \boldsymbol{z}_t) \odot \boldsymbol{h}_{t-1} + \boldsymbol{z}_t \odot \tilde{\boldsymbol{h}}_t$$

where $\boldsymbol{x}_t$ and $\boldsymbol{h}_{t-1}$ are respectively updated by Equation (5) and (6), and $\boldsymbol{V}_z, \boldsymbol{V}_r, \boldsymbol{V}$ are new parameters for masking vector $\boldsymbol{m}_t$.

To validate GRU-D model and demonstrate how it utilizes informative missing patterns, in Figure 4, we show the input decay ($\boldsymbol{\gamma_x}$) plots and hidden decay ($\boldsymbol{\gamma_h}$) histograms for all the variables for predicting mortality on PhysioNet dataset. For input decay, we notice that the decay rate is almost constant for the majority of variables. However, a few variables have large decay which means that the model relies less on the previous observations for prediction. For example, the changes in the variable values of weight, arterial pH, temperature, and respiration rate are known to impact the ICU patients health condition. The hidden decay histograms show the distribution of decay parameters related to each variable. We noticed that the parameters related to variables with smaller missing rate are more spread out. This indicates that the missingness of those variables has more impact on decaying or keeping the hidden states of the models.

Notice that the decay term can be generalized to LSTM straightforwardly. In practical applications, missing values in time series may contain useful information in a variety of ways. A better model should have the flexibility to capture different missing patterns. In order to demonstrate the capacity of our GRU-D model, we discuss some model variations in Appendix A.2.

## 3 EXPERIMENTS

### 3.1 DATASET DESCRIPTIONS AND EXPERIMENTAL DESIGN

We demonstrate the performance of our proposed models on one synthetic and two real-world health-care datasets[1] and compare it to several strong machine learning and deep learning approaches

---

[1]A summary statistics of the three datasets is shown in Appendix A.3.1.

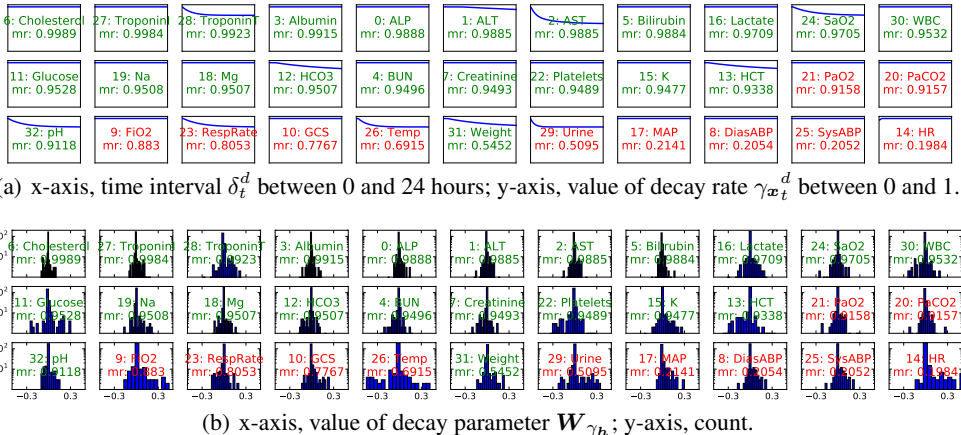

(a) x-axis, time interval $\delta_t^d$ between 0 and 24 hours; y-axis, value of decay rate $\gamma_{\boldsymbol{x}_t}^d$ between 0 and 1.

(b) x-axis, value of decay parameter $\boldsymbol{W}_{\gamma_{\boldsymbol{h}}}$; y-axis, count.

Figure 4: Plots of input decay $\gamma_{\boldsymbol{x}_t}$ (top) and histograms of hidden state decay $\gamma_{\boldsymbol{h}_t}$ (bottom) of all 33 variables in GRU-D model for predicting mortality on PhysioNet dataset. Variables in green are lab measurements; variables in red are vital signs; *mr* refers to missing rate.

in classification tasks. We evaluate our models for different settings such as early prediction and different training sizes and investigate the impact of informative missingness.

**Gesture phase segmentation dataset (Gesture)**  This UCI dataset (Madeo et al., 2013) has multi-variate time series features, regularly sampled and with no missing values, for 5 different gesticulations. We extracted 378 time series and generate 4 synthetic datasets for the purpose of understanding model behaviors with different missing patterns. We treat it as multi-class classification task.

**PhysioNet Challenge 2012 dataset (PhysioNet)**  This dataset, from *PhysioNet Challenge 2012* (Silva et al., 2012), is a publicly available collection of multivariate clinical time series from 8000 intensive care unit (ICU) records. Each record is a multivariate time series of roughly 48 hours and contains 33 variables such as *Albumin, heart-rate, glucose* etc. We used *Training Set A* subset in our experiments since outcomes (such as in-hospital mortality labels) are publicly available only for this subset. We conduct the following two prediction tasks on this dataset: 1) *Mortality task*: Predict whether the patient dies in the hospital. There are 554 patients with positive mortality label. We treat this as a binary classification problem. and 2) *All 4 tasks*: Predict 4 tasks: in-hospital mortality, length-of-stay less than 3 days, whether the patient had a cardiac condition, and whether the patient was recovering from surgery. We treat this as a multi-task classification problem.

**MIMIC-III dataset (MIMIC-III)**  This public dataset (Johnson et al., 2016) has deidentified clinical care data collected at Beth Israel Deaconess Medical Center from 2001 to 2012. It contains over 58,000 hospital admission records. We extracted 99 time series features from 19714 admission records for 4 modalities including input-events (fluids into patient, e.g., insulin), output-events (fluids out of the patient, e.g., urine), lab-events (lab test results, e.g., pH values) and prescription-events (drugs prescribed by doctors, e.g., aspirin). These modalities are known to be extremely useful for monitoring ICU patients. We only use the first 48 hours data after admission from each time series. We perform following two predictive tasks: 1) *Mortality task*: Predict whether the patient dies in the hospital after 48 hours. There are 1716 patients with positive mortality label and we perform binary classification. and 2) *ICD-9 Code tasks*: Predict 20 ICD-9 diagnosis categories (e.g., respiratory system diagnosis) for each admission. We treat this as a multi-task classification problem.

## 3.2 METHODS AND IMPLEMENTATION DETAILS

We categorize all evaluated prediction models into three following groups:

- *Non-RNN Baselines (Non-RNN)*: We evaluate logistic regression (LR), support vector machines (SVM) and Random Forest (RF) which are widely used in health care applications.
- *RNN Baselines (RNN)*: We take GRU-mean, GRU-forward, GRU-simple, and LSTM-mean (LSTM model with mean-imputation on the missing measurements) as RNN baselines.
- *Proposed Methods (Proposed)*: This is our proposed GRU-D model from Section 2.2.

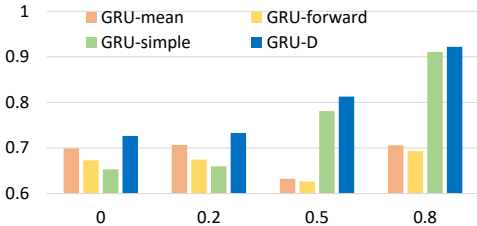

Figure 5: Classification performance on Gesture synthetic datasets. x-axis: average Pearson correlation of variable missing rates and target label in that dataset; y-axis: AUC score.

Table 1: Model performances measured by average AUC score ($mean \pm std$) for multi-task predictions on real datasets. Results on each class are shown in Appendix A.3.3 for reference.

| Models | MIMIC-III ICD-9 20 tasks | PhysioNet All 4 tasks |
|---|---|---|
| GRU-mean | $0.7070 \pm 0.001$ | $0.8099 \pm 0.011$ |
| GRU-forward | $0.7077 \pm 0.001$ | $0.8091 \pm 0.008$ |
| GRU-simple | $0.7105 \pm 0.001$ | $0.8249 \pm 0.010$ |
| GRU-D | $\mathbf{0.7123 \pm 0.003}$ | $\mathbf{0.8370 \pm 0.012}$ |

The non-RNN baselines cannot handle missing data directly. We carefully design experiments for non-RNN models to capture the *informative missingness* as much as possible to have fair comparison with the RNN methods. Since non-RNN models only work with fixed length inputs, we regularly sample the time-series data to get a fixed length input and perform imputation to fill in the missing values. Similar to RNN baselines, we can concatenate the masking vector along with the measurements and feed it to non-RNN models. For PhysioNet dataset, we sample the time series on an hourly basis and propagate measurements forward (or backward) in time to fill gaps. For MIMIC-III dataset, we consider two hourly samples (in the first 48 hours) and do forward (or backward) imputation. Our preliminary experiments showed 2-hourly samples obtains better performance than one-hourly samples for MIMIC-III. We report results for both concatenation of input and masking vectors (i.e., SVM/LR/RF-simple) and only input vector without masking (i.e., SVM/LR/RF-forward). We use the scikit-learn (Pedregosa et al., 2011) for the non-RNN model implementation and tune the parameters by cross-validation. We choose RBF kernel for SVM since it performs better than other kernels.

For RNN models, we use a one layer RNN to model the sequence, and then apply a soft-max regressor on top of the last hidden state $h_T$ to do classification. We use 100 and 64 hidden units in GRU-mean for MIMIC-III and PhysioNet datasets, respectively. All the other RNN models were constructed to have a comparable number of parameters.[2] For GRU-simple, we use mean imputation for input as shown in Equation (1). Batch normalization (Ioffe & Szegedy, 2015) and dropout (Srivastava et al., 2014) of rate 0.5 are applied to the top regressor layer. We train all the RNN models with the Adam optimization method (Kingma & Ba, 2014) and use early stopping to find the best weights on the validation dataset. All the input variables are normalized to be 0 mean and 1 standard deviation. We report the results from 5-fold cross validation in terms of area under the ROC curve (AUC score). We provide more detailed comparisons of RNN baselines and variations in Appendix A.3.4 and evaluations on multilayer RNN models in Appendix A.3.5.

## 3.3 QUANTITATIVE RESULTS

**Exploiting informative missingness on synthetic dataset**   As illustrated in Figure 1, missing patterns can be useful in solving prediction tasks. A robust model should exploit informative missingness properly and avoid inducing nonexistent relations between missingness and predictions. To evaluate the impact of modeling missingness we conduct experiments on the synthetic Gesture datasets. We process the data in 4 different settings with the same missing rate but different correlations between missing rate and the label. A higher correlation implies more informative missingness. Figure 5 shows the AUC score comparison of three GRU baseline models (GRU-mean, GRU-forward, GRU-simple) and the proposed GRU-D. Since GRU-mean and GRU-forward do not utilize any missingness (i.e., masking or time interval), they perform similarly across all 4 settings. GRU-simple and GRU-D benefit from utilizing the missingness, especially when the correlation is high. Our GRU-D achieves the best performance in all settings, while GRU-simple fails when the correlation is low. The results on synthetic datasets demonstrates that our proposed model can model and distinguish useful missing patterns in data properly compared with baselines.

**Prediction task evaluation on real datasets**   We evaluate all methods in Section 3.2 on MIMIC-III and PhysioNet datasets. We noticed that dropout in the recurrent layer helps a lot for all RNN models

---

[2]Appendix A.3.2 compares all GRU models tested in the experiments in terms of model size.

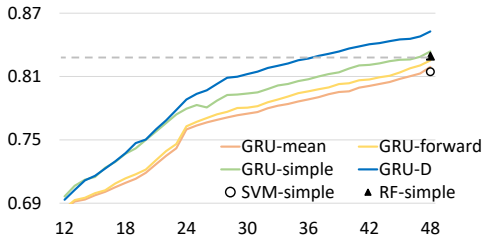

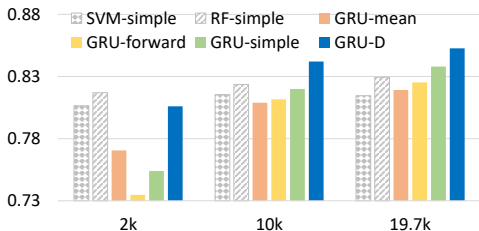

Figure 6: Performance for early predicting mortality on MIMIC-III dataset. x-axis, # of hours after admission; y-axis, AUC score; Dash line, RF-simple results for 48 hours.

Figure 7: Performance for predicting mortality on subsampled MIMIC-III dataset. x-axis, subsampled dataset size; y-axis, AUC score.

on both of the datasets, probably because they contain more input variables and training samples than synthetic dataset. Similar to Gal (2015), we apply dropout rate of 0.3 with same dropout samples at each time step on weights $W, U, V$. Table 2 shows the prediction performance of all the models on mortality task. All models except for random forest improve their performance when they feed missingness indicators along with inputs. The proposed GRU-D achieves the best AUC score on both datasets. We also conduct multi-task classification experiments for *all 4 tasks* on PhysioNet and *20 ICD-9 code tasks* on MIMIC-III using all the GRU models. As shown in Table 1, GRU-D performs best in terms of average AUC score across all tasks and in most of the single tasks.

Table 2: Model performances measured by AUC score ($mean \pm std$) for mortality prediction.

|  | Models | MIMIC-III | PhysioNet |
|---|---|---|---|
| *Non-RNN* | LR-forward | $0.7589 \pm 0.015$ | $0.7423 \pm 0.011$ |
|  | SVM-forward | $0.7908 \pm 0.006$ | $0.8131 \pm 0.018$ |
|  | RF-forward | $0.8293 \pm 0.004$ | $0.8183 \pm 0.015$ |
|  | LR-simple | $0.7715 \pm 0.015$ | $0.7625 \pm 0.004$ |
|  | SVM-simple | $0.8146 \pm 0.008$ | $0.8277 \pm 0.012$ |
|  | RF-simple | $0.8294 \pm 0.007$ | $0.8157 \pm 0.013$ |
| *RNN* | LSTM-mean | $0.8142 \pm 0.014$ | $0.8025 \pm 0.013$ |
|  | GRU-mean | $0.8192 \pm 0.013$ | $0.8195 \pm 0.004$ |
|  | GRU-forward | $0.8252 \pm 0.011$ | $0.8162 \pm 0.014$ |
|  | GRU-simple | $0.8380 \pm 0.008$ | $0.8155 \pm 0.004$ |
| *Proposed* | GRU-D | $\mathbf{0.8527 \pm 0.003}$ | $\mathbf{0.8424 \pm 0.012}$ |

## 3.4 DISCUSSIONS

**Online prediction in early stage** Although our model is trained on the first 48 hours data and makes prediction at the last time step, it can be used directly to make predictions before it sees all the time series and can make predictions on the fly. This is very useful in applications such as health care, where early decision making is beneficial and critical for patient care. Figure 6 shows the online prediction results for MIMIC-III mortality task. As we can see, AUC is around 0.7 at first 12 hours for all the GRU models and it keeps increasing when longer time series is fed into these models. GRU-D and GRU-simple, which explicitly handle missingness, perform consistently better than the other two methods. In addition, GRU-D outperforms GRU-simple when making predictions given time series of more than 24 hours, and has at least 2.5% higher AUC score after 30 hours. This indicates that GRU-D is able to capture and utilize long-range temporal missing patterns. Furthermore, GRU-D achieves similar prediction performance (i.e., same AUC) as best non-RNN baseline model with less time series data. As shown in the figure, GRU-D has same AUC performance at 36 hours as the best non-RNN baseline model (RF-simple) at 48 hours. This 12 hour improvement of GRU-D over non-RNN baseline is highly significant in hospital settings such as ICU where time-saving critical decisions demands accurate early predictions.

**Model Scalability with growing data size** In many practical applications, model scalability with large dataset size is very important. To evaluate the model performance with different training dataset size, we subsample three smaller datasets of 2000 and 10000 admissions from the entire MIMIC-III dataset while keeping the same mortality rate. We compare our proposed models with all GRU baselines and two most competitive non-RNN baselines (SVM-simple, RF-simple) and shows the prediction results in Figure 7. We observe that all models can achieve improved performance given more training samples. However, the improvements of non-RNN baselines are quite limited compared to GRU models, and our GRU-D model achieves the best results on the two larger datasets. These results indicate the performance gap between GRU-D and non-RNN baselines will continue to grow as more data become available.

## 4 SUMMARY

In this paper, we proposed novel GRU-based model to effectively handle missing values in multivariate time series data. Our model captures the *informative missingness* by incorporating masking and time interval directly inside the GRU architecture. Empirical experiments on both synthetic and real-world health care datasets showed promising results and provided insightful findings. In our future work, we will explore deep learning approaches to characterize missing-not-at-random data and we will conduct theoretical analysis to understand the behaviors of existing solutions for missing values.

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

## A APPENDIX

### A.1 INVESTIGATION OF RELATION BETWEEN MISSINGNESS AND LABELS

In many time series applications, the pattern of missing variables in the time series is often informative and useful for prediction tasks. Here, we empirically confirm this claim on real health care dataset by investigating the correlation between the missingness and prediction labels (mortality and ICD-9 diagnosis categories). We denote the missing rate for a variable $d$ as $p_{\boldsymbol{X}}^d$ and calculate it by $p_{\boldsymbol{X}}^d = 1 - \frac{1}{T}\sum_{t=1}^{T} m_t^d$. Note that $p_{\boldsymbol{X}}^d$ is dependent on mask vector ($m_t^d$) and number of time steps $T$. For each prediction task, we compute the Pearson correlation coefficient between $p_{\boldsymbol{X}}^d$ and label $\ell$ across all the time series. As shown in Figure 1, we observe that on MIMIC-III dataset the missing rates with low rate values are usually highly (either positive or negative) correlated with the labels. The distinct correlation between missingness and labels demonstrates usefulness of missingness patterns in solving prediction tasks.

### A.2 GRU-D MODEL VARIATIONS

In this section, we will discuss some variations of GRU-D model, and also compare some related RNN models which are used for time series with missing data with the proposed model.

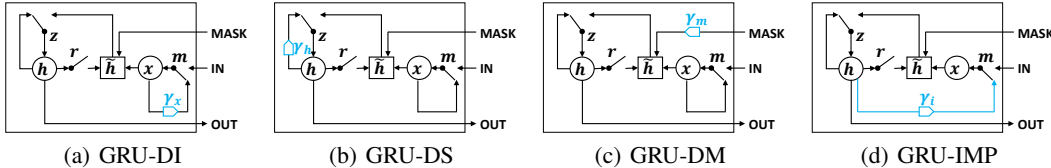

|        (a) GRU-DI        |        (b) GRU-DS        |        (c) GRU-DM        |        (d) GRU-IMP        |

Figure 8: Graphical illustrations of variations of proposed GRU models.

#### A.2.1 GRU MODEL WITH DIFFERENT TRAINABLE DECAYS

The proposed GRU-D applies trainable decays on both input and hidden state transitions in order to capture the temporal missing patterns explicitly. This decay idea can be straightforwardly generated to other parts inside the GRU models separately or jointly, given different assumptions on the impact of missingness. As comparisons, we also describe and evaluate several modifications of GRU-D model.

**GRU-DI** (Figure 8(a)) and **GRU-DS** (Figure 8(b)) decay only the input and only the hidden state by Equation (5) and (6), respectively. They can be considered as two simplified models of the proposed GRU-D. GRU-DI aims at capturing direct impact of missing values in the data, while GRU-DS captures more indirect impact of missingness. Another intuition comes from this perspective: if an input variable is just missing, we should pay more attention to this missingness; however, if an variable has been missing for a long time and keeps missing, the missingness becomes less important. We can utilize this assumption by decaying the masking. This brings us the model **GRU-DM** shown in Figure 8(c), where we replace the masking $m_t^d$ fed into GRU-D in by

$$m_t^d \leftarrow m_t^d + (1 - m_t^d)\gamma_{\boldsymbol{m}_t}^d(1 - m_t^d) = m_t^d + (1 - m_t^d)\gamma_{\boldsymbol{m}_t}^d \tag{7}$$

where the equality holds since $m_t^d$ is either 0 or 1. We decay the masking for each variable independently from others by constraining $\boldsymbol{W}_{\gamma_{\boldsymbol{m}}}$ to be diagonal.

#### A.2.2 GRU-IMP: GOAL-ORIENTED IMPUTATION MODEL

We may alternatively let the GRU-RNN predict the missing values in the next timestep on its own. When missing values occur only during test time, we simply train the model to predict the measurement vector of the next time step as a language model (Mikolov et al., 2010) and use it to fill the missing values during test time. This is unfortunately not applicable for some time series applications such as in health care domains, which also have missing data during training.

Instead, we propose goal-oriented imputation model here called **GRU-IMP**, and view missing values as latent variables in a probabilistic graphical model. Given a timeseries $\boldsymbol{X}$, we denote all the missing variables by $\mathcal{M}_{\boldsymbol{X}}$ and all the observed ones by $\mathcal{O}_{\boldsymbol{X}}$. Then, training a time-series classifier with missing variables becomes equivalent to maximizing the marginalized log-conditional probability of a correct label $l$, i.e., $\log p(l|\mathcal{O}_{\boldsymbol{X}})$.

The exact marginalized log-conditional probability is however intractable to compute, and we instead maximize its lowerbound:

$$\log p(l|\mathcal{O}_{\boldsymbol{X}}) = \log \sum_{\mathcal{M}_{\boldsymbol{X}}} p(l|\mathcal{M}_{\boldsymbol{X}}, \mathcal{O}_{\boldsymbol{X}}) \, p(\mathcal{M}_{\boldsymbol{X}}|\mathcal{O}_{\boldsymbol{X}}) \geq \mathbb{E}_{\mathcal{M}_{\boldsymbol{X}} \sim p(\mathcal{M}_{\boldsymbol{X}}|\mathcal{O}_{\boldsymbol{X}})} \log p(l|\mathcal{M}_{\boldsymbol{X}}, \mathcal{O}_{\boldsymbol{X}})$$

where we assume the distribution over the missing variables at each time step is only conditioned on all the previous observations:

$$p(\mathcal{M}_{\boldsymbol{X}}|\mathcal{O}_{\boldsymbol{X}}) = \prod_{t=1}^{T} \prod_{\substack{1 \leq d \leq D}}^{m_t^d = 1} p(x_t^d|\boldsymbol{x}_{1:(t-1)}, \boldsymbol{m}_{1:(t-1)}, \boldsymbol{\delta}_{1:(t-1)}) \tag{8}$$

Although this lowerbound is still intractable to compute exactly, we can approximate it by Monte Carlo method, which amounts to sampling the missing variables at each time as the RNN reads the input sequence from the beginning to the end, such that

$$x_t^d \leftarrow m_t^d x_t^d + (1 - m_t^d)\tilde{x}_t^d \tag{9}$$

where $\tilde{\boldsymbol{x}}_t \sim x_t^d|\boldsymbol{x}_{1:(t-1)}, \boldsymbol{m}_{1:(t-1)}, \boldsymbol{\delta}_{1:(t-1)}$.

By further assuming that $\tilde{\boldsymbol{x}}_t \sim \mathcal{N}\left(\boldsymbol{\mu}_t, \boldsymbol{\sigma}_t^2\right)$, $\boldsymbol{\mu}_t = \boldsymbol{\gamma}_t \odot \left(\boldsymbol{W}_x \boldsymbol{h}_{t-1} + \boldsymbol{b}_x\right)$ and $\boldsymbol{\sigma}_t = \boldsymbol{1}$, we can use a reparametrization technique widely used in stochastic variational inference (Kingma & Welling, 2013; Rezende et al., 2014) to estimate the gradient of the lowerbound efficiently. During the test time, we simply use the mean of the missing variable, i.e., $\tilde{\boldsymbol{x}}_t = \boldsymbol{\mu}_t$, as we have not seen any improvement from Monte Carlo approximation in our preliminary experiments. We view this approach as a goal-oriented imputation method and show its structure in Figure 8(d). The whole model is trained to minimize the classification cross-entropy error $\ell_{log\_loss}$ and we take the negative log likelihood of the observed values as a regularizer.

$$\ell = \ell_{log\_loss} + \lambda \frac{1}{N} \sum_{n=1}^{N} \frac{1}{T_n} \sum_{t=1}^{T_n} \frac{\sum_{d=1}^{D} m_t^d \cdot \log p(x_t^d|\mu_t^d, \sigma_t^d)}{\sum_{d=1}^{D} m_t^d} \tag{10}$$

### A.3 SUPPLEMENTARY EXPERIMENT DETAILS

#### A.3.1 DATA STATISTICS

For each of the three datasets used in our experiments, we list the number of samples, the number of input variables, the mean and max number of time steps for all the samples, and the mean of all the variable missing rates in Table 3.

Table 3: Dataset statistics.

|  | MIMIC-III | PhysioNet2012 | Gesture |
|---|---|---|---|
| # of samples ($N$) | 19714 | 4000 | 378 |
| # of variables ($D$) | 99 | 33 | 23 |
| Mean of # of time steps | 35.89 | 68.91 | 21.42 |
| Maximum of # of time steps | 150 | 155 | 31 |
| Mean of variable missing rate | 0.9621 | 0.8225 | N/A |

A.3.2   GRU MODEL SIZE COMPARISON

In order to fairly compare the capacity of all GRU-RNN models, we build each model in proper size so they share similar number of parameters. Table 4 shows the statistics of all GRU-based models for on three datasets. We show the statistics for mortality prediction on the two real datasets, and it's almost the same for multi-task classifications tasks on these datasets. In addition, having comparable number of parameters also makes all the models have number of iterations and training time close in the same scale in all the experiments.

Table 4: Comparison of GRU model size in our experiments. *Size* refers to the number of hidden states ($h$) in GRU .

| Models | Gesture 18 input variables | | MIMIC-III 99 input variables | | PhysioNet 33 input variables | |
|---|---|---|---|---|---|---|
| | *Size* | # of parameters | *Size* | # of parameters | *Size* | # of parameters |
| GRU-mean&forward | 64 | 16281 | 100 | 60105 | 64 | 18885 |
| GRU-simple | 50 | 16025 | 56 | 59533 | 43 | 18495 |
| GRU-D | 55 | 16561 | 67 | 60436 | 49 | 18838 |

A.3.3   MULTI-TASK PREDICTION DETAILS

The RNN models for multi-task learning with $m$ tasks is almost the same as that for binary classification, except that 1) the soft-max prediction layer is replaced by a fully connected layer with $n$ sigmoid logistic functions, and 2) a data-driven prior regularizer (Che et al., 2015), parameterized by comorbidity (co-occurrence) counts in training data, is applied to the prediction layer to improve the classification performance. We show the AUC scores for predicting 20 ICD-9 diagnosis categories on MIMIC-III dataset in Figure 9, and all 4 tasks on PhysioNet dataset in Figure 10. The proposed GRU-D achieves the best average AUC score on both datasets and wins 11 of the 20 ICD-9 prediction tasks.

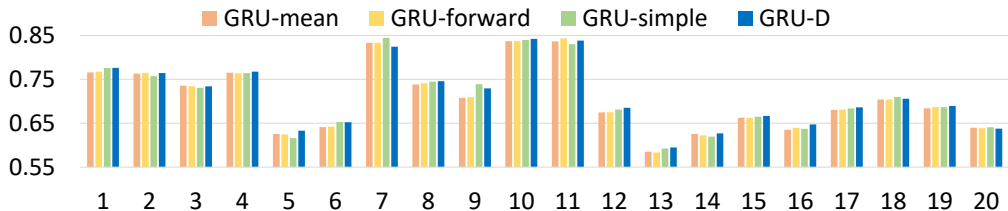

Figure 9: Performance for predicting 20 ICD-9 diagnosis categories on MIMIC-III dataset. x-axis, ICD-9 diagnosis category id; y-axis, AUC score.

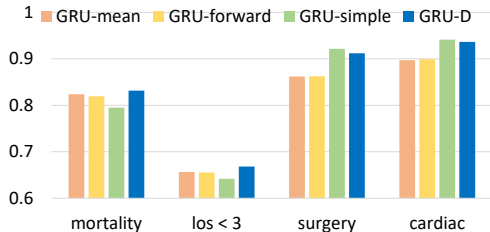

Figure 10: Performance for predicting all 4 tasks on PhysioNet dataset. *mortality*, in-hospital mortality; *los*< 3, length-of-stay less than 3 days; *surgery*, whether the patient was recovering from surgery; *cardiac*, whether the patient had a cardiac condition; y-axis, AUC score.

A.3.4 EMPIRICAL COMPARISON OF MODEL VARIATIONS

As a thorough empirical comparison, we test all GRU model variations mentioned in Appendix A.2 along with the proposed GRU-D. These include 1) 4 models with trainable decays (GRU-DI, GRU-DS, GRU-DM, GRU-IMP), and 2) two models simplified from GRU-simple (interval only and masking only). The results are shown in Table 5. As we can see, GRU-D performs best among these models.

Table 5: Model performances of GRU variations measured by AUC score ($mean \pm std$) for mortality prediction.

|  | Models | MIMIC-III | PhysioNet |
|---|---|---|---|
|  | GRU-simple (masking only) | $0.8367 \pm 0.009$ | $0.8226 \pm 0.010$ |
| *Baselines* | GRU-simple (interval only) | $0.8266 \pm 0.009$ | $0.8125 \pm 0.005$ |
|  | **GRU-simple** | $0.8380 \pm 0.008$ | $0.8155 \pm 0.004$ |
|  | GRU-DI | $0.8345 \pm 0.006$ | $0.8328 \pm 0.008$ |
|  | GRU-DS | $0.8425 \pm 0.006$ | $0.8241 \pm 0.009$ |
| *Proposed* | GRU-DM | $0.8342 \pm 0.005$ | $0.8248 \pm 0.009$ |
|  | GRU-IMP | $0.8248 \pm 0.010$ | $0.8231 \pm 0.005$ |
|  | **GRU-D** | $\mathbf{0.8527 \pm 0.003}$ | $\mathbf{0.8424 \pm 0.012}$ |

A.3.5 EVALUATION ON MULTI-LAYER RNNS

We also conducted experiments on 2-layer RNN models to demonstrate the superiority of our proposed GRU-D models can be generalized to multi-layer RNNs. For all baseline and proposed GRU models, we add one standard GRU layer on top of the baseline or proposed GRU layer. We tested models both with similar number of parameters to single layer models and with more parameters. As shown in Table 6 and 7, our GRU-D model consistently outperforms other baselines in all cases, and models with moderate size perform as good as larger models with more parameters. Compared with 1-layer RNNs, all models with deeper structures perform much better on the large MIMIC-III dataset but no better on the relative small PhysioNet dataset.

Table 6: Comparison of multi-layer GRU models for mortality prediction on PhysioNet dataset. *Size* refers to the numbers of hidden states of 2 GRU layers.

|  | Models | PhysioNet | | |
|---|---|---|---|---|
|  |  | *Size* | # of params. | AUC score |
|  | GRU-mean | $40, 40$ | 18643 | $0.8157 \pm 0.008$ |
| *Similar size* | GRU-forward | $40, 40$ | 18643 | $0.8205 \pm 0.008$ |
|  | GRU-simple | $32, 32$ | 18947 | $0.8159 \pm 0.007$ |
|  | **GRU-D** | $34, 34$ | 18599 | $\mathbf{0.8420 \pm 0.009}$ |
|  | GRU-mean | $64, 64$ | 43651 | $0.8199 \pm 0.002$ |
| *Larger size* | GRU-forward | $64, 64$ | 43651 | $0.8112 \pm 0.035$ |
|  | GRU-simple | $43, 64$ | 39250 | $0.8208 \pm 0.009$ |
|  | **GRU-D** | $49, 64$ | 40739 | $\mathbf{0.8363 \pm 0.013}$ |

Table 7: Comparison of multi-layer GRU models for mortality prediction on MIMIC-III dataset. *Size* refers to the numbers of hidden states of 2 GRU layers.

| Models | | MIMIC-III | | |
|---|---|---|---|---|
| | | *Size* | # of params. | AUC score |
| *Similar size* | GRU-mean | $66, 66$ | $59271$ | $0.9538 \pm 0.005$ |
| | GRU-forward | $66, 66$ | $59271$ | $0.9441 \pm 0.005$ |
| | GRU-simple | $46, 46$ | $60355$ | $0.9527 \pm 0.005$ |
| | **GRU-D** | $52, 52$ | $60989$ | $\mathbf{0.9606 \pm 0.002}$ |
| *Larger size* | GRU-mean | $100, 64$ | $91747$ | $0.9523 \pm 0.006$ |
| | GRU-forward | $100, 64$ | $91747$ | $0.9443 \pm 0.003$ |
| | GRU-simple | $56, 64$ | $82771$ | $0.9520 \pm 0.003$ |
| | **GRU-D** | $67, 64$ | $85775$ | $\mathbf{0.9604 \pm 0.003}$ |
| | GRU-mean | $100, 128$ | $148067$ | $0.9539 \pm 0.006$ |
| | GRU-forward | $100, 128$ | $148067$ | $0.9457 \pm 0.005$ |
| | GRU-simple | $56, 128$ | $130643$ | $0.9523 \pm 0.003$ |
| | **GRU-D** | $67, 128$ | $135759$ | $\mathbf{0.9618 \pm 0.002}$ |

