# Peer review of "Recurrent Neural Networks for Multivariate Time Series with Missing Values"

_ICLR 2017 — rejected_

[Public Comment · Alexey Romanov · 11 Nov 2016]
**A couple of questions**

Thank you for a very interesting work! 


I have a couple of questions:

1) Lipton at al. (2016) achieve the best result with zero filling and indicators. As I understood from the equation 3 and the following description, you did not experiment with zero filling and used either forward or mean imputation. Is it correct?

2) It will be interesting to see not only AUC but also Sensitivity and Positive Predictivity, as well as min(Se, +P) since that was the official scoring metric in the PhysioNet Challenge 2012 (although we cannot compare these score directly, obviously)

3) How did you combat such high class imbalance in case of mortality prediction?

[Official Review · AnonReviewer2 · rating 6 · confidence 3 · 15 Dec 2016]
**No Title**

The authors propose a RNN-method for time-series classification with missing values, that can make use of potential information in missing values. It is based on a simple linear imputation of missing values with learnable parameters. Furthermore, time-intervals between missing values are computed and used to scale the RNN computation downstream. The authors demonstrate that their method outperforms reasonable baselines on (small to mid-sized) real world datasets. The paper is clearly written.
IMO the authors propose a reasonable approach for dealing with missing values for their intended application domain, where data is not abundant and requires smallish models. I’m somewhat sceptical if the benefits would carry over to big datasets, where more general, less handcrafted multi-layer RNNs are an option.

[Official Review · AnonReviewer5 · rating 6 · confidence 4 · 17 Dec 2016 (modified: 19 Jan 2017)]
**Interesting model, overreaching conclusions**

This paper presents a modified gated RNN caled GRU-D that deals with time series which display a lot of missing values in their input. They work on two fronts. The first deals with the missing inputs directly by using a learned convex combination of the previous available value (forward imputation) and the mean value (mean imputation). The second includes dampening the recurrent layer not unlike a second reset gate, but parametrized according to the time elapsed since the last available value of each attributes.

Positives
------------
- Clear definition of the task (handling missing values for classification of time series)
- Many interesting baselines to test the new model against.
- The model presented deals with the missing values in a novel, ML-type way (learn new dampening parameters).
- The extensive tests done on the datasets is probably the greatest asset of this paper.

Negatives
-------------
- The paper could use some double checking for typos.
- The Section A.2.3 really belongs in the main article as it deals with important related works. Swap it with the imprecise diagrams of the model if you need space.
- No mention of any methods from the statistics litterature.

Here are the two main points of this review that informs my decision:

1. The results, while promising, are below expectations. The paper hasn’t been able to convince me that GRU-simple (without intervals) isn’t just as well-suited for the task of handling missing inputs as GRU-D. In the main paper, GRU-simple is presented as the main baseline. Yet, it includes a lot of extraneous parameters (the intervals) that, according to Table 5, probably hurts the model more than it helps it. Having a third of it’s parameters being of dubious value, it brings the question of the fairness of the comparison done in the main paper, especially since in the one table where GRU-simple (without intervals) is present, GRU-D doesn’t significantly outperforms it.

2. My second concern, and biggest, is with some claims that are peppered through the paper. The first is about the relationship with the presence rate of data in the dataset and the diagnostics. I might be wrong, but that only indicates that the doctor in charge of that patient requested the relevant analyses be done according to the patient’s condition. That would mean that an expert system based on this data would always seem to be one step behind. 
The second claim is the last sentence of the introduction, which sets huge expectations that were not met by the paper. Another is that “simply concatenating masking and time interval vectors fails to exploit the temporal structure of missing values” is unsubstantiated and actually disproven later in the paper. 
Yet another is the conclusion that since GRU models displayed the best improvement between a subsample of the dataset and the whole of it means that the improvement is going to continue to grow as more data is added. This fails to consider that non-GRU models actually started with much better results than most GRU ones. 
Lastly is their claim to capture informative missingness by incorporating masking and time intervals directly inside the GRU architecture. While the authors did make these changes, the fact that they also concatenate the mask to the input, just like GRU-simple (without intervals), leads me to question the actual improvement made by GRU-D. 

Given that, while I find that the work that has been put into the paper is above average, I wouldn’t accept that paper without a reframing of the findings and a better focus on the real contribution of this paper, which I believe is the novel way to parametrize the choice of imputation method.

[Official Review · AnonReviewer4 · rating 5 · confidence 3 · 17 Dec 2016]
**No Title**

This paper proposed a way to deal with supervised multivariate time series tasks involving missing values. The high level idea is still using the recurrent neural network (specifically, GRU in this paper) to do sequence supervised learning, e.g., classification, but modifications have been made to the input and hidden layers of RNNs to tackle the missing value problem. 

pros: 
1) the insight of utilizing missing value is critical. the observation of decaying effect in the healthcare application is also interesting;
2) the experiment seems to be solid; the baseline algorithms and analysis of results are also done properly. 

cons:
1) the novelty of this work is not enough. Adding a decaying smooth factor to input and hidden layers seems to be the main modification of the architecture. 
2) the datasets used in this paper are small. 
3) the decaying effect might not be able to generalize to other domains.

[Final Decision · Program Chairs · 06 Feb 2017]
**ICLR committee final decision**

This paper presents a modification of GRU-RNNs to handle missing data explicitly, allowing them to exploit data not missing at random. The method is presented clearly enough, but the reviewers felt that the claims were overreaching. It's also unsatisfying that the method depends on specific modifications of RNN architectures for a particular domain, instead of being a more general approach.